# Estimating maternity ward birth attendant time use in India: a microcosting study

Katherine T Lofgren [ID],[1] Lauren Bobanski,[2] Danielle E Tuller,[2] Vinay P Singh,[3,4] Megan Marx Delaney,[2] Amanda Jurczak,[2] Meera Ragavan,[5] Tapan Kalita,[6] Ami Karlage,[2] Stephen Charles Resch [ID],[7] Katherine E A Semrau [ID] [2,8,9]

For numbered affiliations see end of article.

**Correspondence to**
Dr Katherine T Lofgren;
kate.lofgren@gmail.com

## ABSTRACT

**Objectives** Despite global concern over the quality of maternal care, little is known about the time requirements to complete the essential birth practices. Using three microcosting data collection methods within the BetterBirth trial, we aimed to assess time use and the specific time requirements to incorporate the WHO Safe Childbirth Checklist into clinical practice.

**Setting** We collected detailed survey data on birth attendant time use within the BetterBirth trial in Uttar Pradesh, India. The BetterBirth trial tested whether the peer-coaching-based implementation of the WHO Checklist was effective in improving the quality of facility-based childbirth care.

**Participants** We collected measurements of time to completion for 18 essential birth practices from July 2016 through October 2016 across 10 facilities in five districts (1559 total timed observations). An anonymous survey asked about the impact of the WHO Checklist on birth attendants at every intervention facility (15 facilities, 83 respondents) in the Lucknow hub. Additionally, data collectors visited facilities to conduct a census of patients and birth attendants across 20 facilities in seven districts between June 2016 and November 2016 (six hundred and ten 2-hour facility observations).

**Primary and secondary outcome measures** The primary outcome measure of this study is the per cent of staff time required to complete the essential birth practices included in the WHO Checklist.

**Results** When birth attendants were timed, we found practices were completed rapidly (18 s to 2 min). As the patient load increased, time dedicated to clinical care increased but remained low relative to administrative and downtime. On average, WHO Checklist clinical care accounted for less than 7% of birth attendant time use per hour.

**Conclusions** We did not find that a coaching-based implementation of the WHO Checklist was a burden on birth attendant's time use. However, questions remain regarding the performance quality of practices and how to accurately capture and interpret idle and break time.

**Trial registration number** NCT02148952.

## INTRODUCTION

Remarkable achievements have been made in the reduction of maternal and neonatal

### Strengths and limitations of this study

► Three distinct time-use capture methods were used to estimate the time requirements of a coaching-based implementation of the WHO Safe Childbirth Checklist.

► Both stopwatch and birth attendant reported time-to-complete individual evidence-based practices were used to estimate time-to-complete essential birth practices.

► A census of birth attendants and patients in combination with work sampling data on birth attendant time use was used to estimate the per cent of a staffing hour spent on general task categories including administrative duties, Checklist-based practices, clinical non-Checklist and downtime.

► As this work was embedded within the BetterBirth trial, only a subset of treatment facilities were sampled and we are not able to compare across treatment and control facilities or conduct subanalyses by facility type.

► Further work is needed to differentiate true downtime from watchful waiting as this study does not include breakdowns within the broad category of downtime.

mortality globally.[1 2] One of the primary achievements of the Millennium Development Goals era was increased rate of facility-based childbirth.[3] However, evidence suggests increased coverage of services does not necessarily lead to mortality and morbidity reductions.[4 5] A large portion of stillbirths and maternal and neonatal deaths remain preventable with timely, high-quality care.[6–8] As low and middle-income countries (LMICs) continue to expand access to services, ensuring patients receive high-quality, evidence-based clinical care is essential for continued progress in population health.[9]

The use of evidence-based care in labour and delivery facilities remains low.[10 11] Even when women reach a facility in a timely manner, without adequate and appropriate

treatment, preventable deaths occur.[12] Many interventions seek to improve the quality of care in LMIC health systems by increasing the number of essential birth practices performed for each labouring mother.[9] The WHO Safe Childbirth Checklist (Checklist) is one such effort.[13] The Checklist is a clinical care aid that synthesises and prioritises evidence-based essential birth practices (practices) from admission to discharge in order to increase the number of practices—like handwashing, checking the mother for bleeding or discussing family planning—performed by birth attendants (BAs) at the point of care. Defining essential practices and creating mechanisms like the Checklist for clinical staff to consistently implement those practices has been successful across a diverse set of clinical contexts in both high-income and low-income settings.[14–16]

One complicating factor in quality improvement efforts targeting labour and delivery wards specifically is staff time availability. Across health systems, BAs often report feeling overwhelmed and busy.[17] Additionally, staffing shortages are a known barrier to timely, high-quality clinical care.[18] With any quality improvement intervention, clinical care may increase staffing time demands or replace existing low-value activities. The implementation of quality improvement interventions requires understanding existing staff time capacity at baseline and how staff time use changes after implementation.

The BetterBirth trial was a matched-pair, cluster-randomised, controlled trial of a coaching-based implementation of the Checklist in Uttar Pradesh, India, to test the effect of the intervention on a composite outcome of perinatal mortality, maternal mortality or maternal severe complication within 7 days of giving birth.[19 20] Embedded in the BetterBirth trial, we conducted data collection to measure the time demands of the Checklist practices, with the primary intent of informing a cost-effectiveness analysis (CEA) of the BetterBirth trial. When the main outcome of the BetterBirth trial was a null effect on maternal and neonatal mortality and morbidity, the CEA was rendered irrelevant. However, concerns remained about the possibility that the Checklist introduced a significant time burden on BAs. Prior to the implementation of the BetterBirth trial, BAs often reported feeling overwhelmed and busy. As a result, we used the collected data to answer the following questions:

1. What is the time burden of the practices included in the Checklist?
2. Do BAs perceive the Checklist as a significant stress or time burden?
3. How does BA time use change as their patient load increases?

## METHODS
### Study setting
The BetterBirth trial was a peer-coaching-based implementation of the Checklist in Uttar Pradesh, India. The matched-pair, cluster-randomised, controlled trial

randomised the BetterBirth trial across 120 facilities (60 control, 60 treatment) with a study population of women and their newborns, the BAs providing care. Study facilities had more than 1000 deliveries per year and minimum of four labour and delivery staff. The study protocol and results have been published and include further details on the study population, design and methods used to test the primary outcome of interest, maternal and perinatal mortality and morbidity outcomes.[19 20]

This paper details three time-use data collection methods to triangulate the time burden of the Checklist practices within the broader time demands on BAs within the BetterBirth trial.[21] Data collectors (n=16) were junior nurses who received training and supportive supervision for data quality assurance across all three data collection methods (each described in more detail in subsections below). We captured 18 specific Checklist practices (online supplemental table A1) as well as non-Checklist clinical care, administrative duties and break/downtime. Although the intention was to distinguish between a scheduled break and non-scheduled downtime, efforts to delineate between these two activities by data collectors were difficult in practice. For the purposes of this paper, 'downtime' refers to a mix of scheduled breaks as well as idle time for other reasons, such as no patients or watchful waiting during clinical care. We first measured time to completion for 18 practices via direct BA observation during clinical practice (time demand). We then surveyed BAs about their experience during the BetterBirth trial (perceived time demand). Finally, we visited facilities and conducted both a census of births as well as observing clinical care activities at regular intervals (BA time use).

### The time demand of Checklist practices
We collected measurements of Checklist practice time to completion for 18 practices over a 4-month period from July 2016 through October 2016 across 10 facilities in five districts. Data collectors visited each facility two to three times per week for 8-hour shifts between 07:00 and 15:00 or 11:00 and 19:00. If available, a second data collector or a supervisor performed data quality assurance activities. Time-to-complete tasks were assessed by the data collectors with stopwatches, recorded on paper (online supplemental table A2) and transferred to an Excel spreadsheet. The time measurements were used to estimate the time required to complete each Checklist practice.

### Perceived time demand of the Checklist by BAs
We also surveyed BAs on their time burden perceptions. The anonymous survey asked general questions about the impact of the Checklist on the daily routines and workloads of BAs (83 respondents) at every intervention facility (15 facilities) in the Lucknow hub (the cost-effectiveness data collection survey region with 30 total facilities) from June to July 2016. All staff working at the facility on the day of data collection were provided the survey and could answer anonymously. The survey also asked respondents

to rank the top three most time-consuming items on the Checklist and estimate the time required to complete those tasks. The specific time estimates for Checklist practices were used to supplement and compare with the stopwatch time measurements.

## BA time use in the labour and delivery ward

How providers use their time depends on both the patient demand and the number of BAs on duty. To estimate the patient demand and healthcare labour supply, data collectors visited facilities to conduct a census of patients and BAs every 2 hours, recording the results on a paper form (online supplemental table A3). Observations were taken across 20 facilities in seven districts from June 2016 to November 2016. These data were used to calculate the average number of patients per BA at given facilities and times of day.

In addition to the census, we also observed BAs conducting regular care (a work sampling approach) to capture the proportion of time spent on various types of clinical and non-clinical work.[22] A data collector visited a facility and, for each hour observed, recorded the type of activity the BA was engaged in at prespecified 2 min intervals on a paper form (online supplemental table A4). For example, if at 11:00 the BA was using a neonatal bag and mask, the data collector recorded that activity. At 11:02, the data collector would again record what the BA was doing; in some cases, she might still be using a neonatal bag and mask, while in other cases, there may be a new activity listed such as non-Checklist direct patient care. This type of data provides estimates of proportional time spent on various activities but does not directly estimate the time required for specific tasks.

If there were at least two BAs on duty at the same time, observations alternated between two BAs. For example, the 11:00 observation would pertain to BA1 while the 11:02 observation would pertain to BA2, alternating back and forth throughout the hour. If only one BA was available for observation, an observation was taken every 2 min for their work. We calculate the proportion of each BA's time spent in different general activity categories to estimate the overall time use in given facility hours (online supplemental table A5 maps specific activities to general categories).

## Public involvement in research

Patient and provider representatives worked with us to refine the Checklist when it was originally designed in 2009. The BetterBirth trial study research question and design did not have direct patient involvement, but did have a scientific advisory committee that included clinicians, researchers and government officials who work in the same area. We did modify the dissemination plan based on feedback from providers and government partners from each participating facility/district. Further, we published a report for wider dissemination at betterbirth.ariadnelabs.org

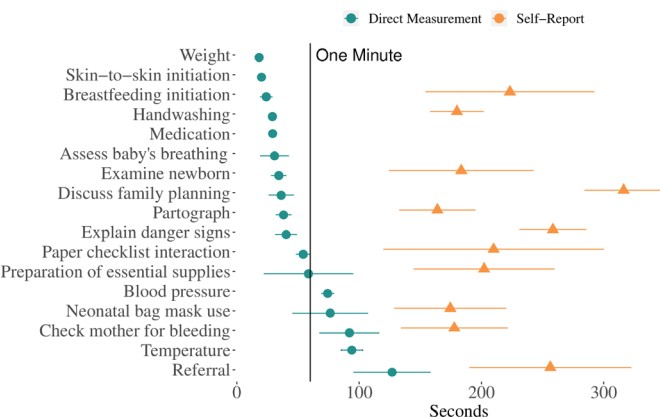

**Figure 1** Time-to-complete specific Checklist-related tasks. Tasks with two or fewer observations have been excluded from this graph. The underlying data for this graph is available in online supplemental table A6.

## RESULTS

### The time demand of Checklist practices

Across all Checklist practices, a total of 1559 practices were directly timed from 35 unique BAs across 10 facilities during clinical care (see online supplemental table A6 for practice-specific sample sizes). The administration of medication (n=419) and handwashing (n=208) were the most frequently observed direct measurements, while referrals (n=21) and the assessment of the baby's breathing (n=9) had the fewest recorded observations. Directly measured task times revealed a pattern of rapid time-to-complete practices on the Checklist. When Checklist practices were directly measured using stopwatches, the average time to complete the task ranged from 127 s (a referral) to 18 s (weighing the baby). Tasks like breastfeeding initiation and discussing family planning that require conversations and (potentially) complex patient–BA interactions both took less than 1 min on average (dots in figure 1, online supplemental table A6). Over 70% (n=12 out of 18 practices) of the average time-to-complete measurements were less than 1 min.

### Perceived time demand of the Checklist by BAs

Across 15 facilities, there were 83 total respondents to the survey. The majority of BAs responded that the Checklist made their jobs easier (96%; n=80). When BAs were asked if the Checklist took away from non-Checklist activities, only 17% of responders felt other clinical duties were rushed (n=11) or their workday was prolonged (n=3).

Respondents were asked to rank the three most time-consuming Checklist practices and estimate the time required to complete those three tasks. Discussing family planning was the most frequently reported time-consuming activity (ranked 1 by 49% of BAs, online supplemental table A7). All tasks were estimated by BAs to take less than 5:07 min on average. The self-reported task times were longer than the direct measurements, particularly in discussion-based practices like explaining danger signs and discussion of family planning (figure 1).

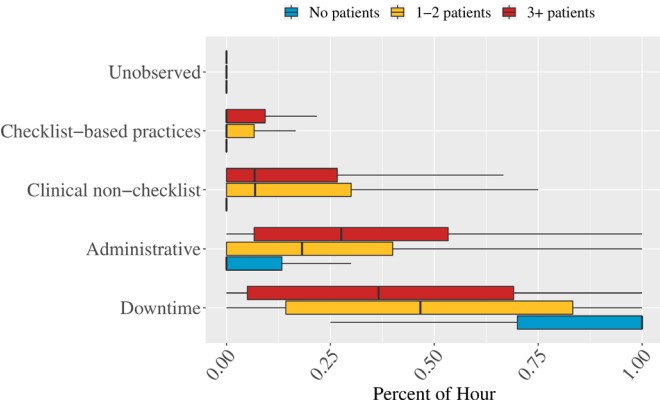

**Figure 2** Birth attendant tasks stratified by patient load per provider. Coloured bar regions represent the IQR.

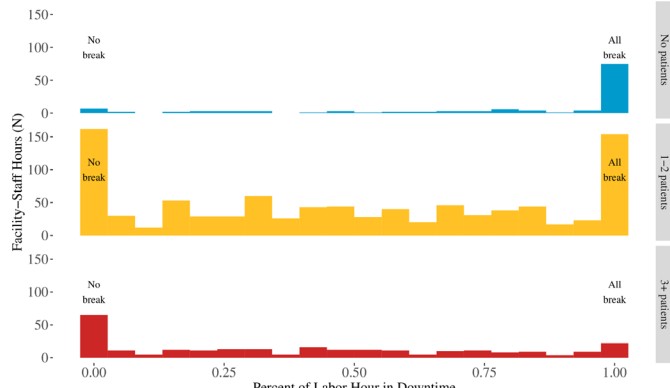

**Figure 3** Per cent of facility staff hours recorded as downtime by patient load per birth attendant.

## BA time use in the labour and delivery ward

BA time use incorporates data from the work sampling time-use data collection and the facility BA and patient census. In total, six hundred and ten 2-hour facility periods were recorded for the patient census and 27 768 individual task observations were recorded in our work sampling survey. Within the hours of data collection (07:00–19:00), we found relatively constant median patient load at 1.4 patients per BA with large variability in the potential patient load for any given facility hour (0–8 patients-per-BA observed range) (online supplemental figure A1).

Clinical care (both non-Checklist and Checklist) was 21% of the average facility staff hour. As patients per BA increased, so did clinical care and administrative duties. When there were no patients, BAs spent the majority of their time in downtime (79% of time) or conducting administrative tasks (15% of time) and 1% of time on Checklist clinical care. Once the patient load increased to one to two patients per BA, BA time use shifted towards clinical care (24% of BA time; 5% Checklist specific) as well as administrative tasks (25% up from 15% with no patients). At three or more patients per BA, the Checklist accounted for 7% of BA time (out of a total 26% of the hour spent on clinical care). However, even at high patient loads (3+), the most common time use, on average, was still recorded as downtime (40% in downtime compared with 26% in clinical care; figure 2). Sample size breakdowns for individual work sampling observations by patient load and task type are available in online supplemental table A8.

However, the average BA downtime is a misleading statistic. When the full distribution of BA downtime by facility BA hour is graphed, there is clear heterogeneity in the distribution that is not captured by summary measures like the mean or median per cent of the hour spent in downtime. Particularly for the one to two patient categories, there is a clear bimodal trend with BAs spending the majority of staff hours either completely in downtime or without any downtime. Similarly, for the 0 and 3+ patient categories, the distributions are highly skewed to all downtime (0 patient) and no downtime (3+ patients).

Taken across all these categories, summary statistics mask the extreme downtime dichotomy experienced in practice by BAs (figure 3, online supplemental figure A2). Sample size breakdowns for individual work sampling observations by patient load and task type are available in online supplemental table A9.

## DISCUSSION

The time demand on BAs is an important piece of the maternal and newborn quality-of-care puzzle. Quality improvement efforts inherently require staff time to shift away from existing time uses and towards evidence-based practices such as those included in the Checklist. Using three different data collection efforts, we found that the Checklist practices were not an undue time burden on BAs. However, based on our data, we are not confident that practices were performed at sufficient quality. Further, our results show a high proportion of ambiguously measured downtime, a lesson to learn from in future studies to differentiate watchful waiting from true downtime. Concerns about the quality of care provided are consistent with the overall BetterBirth trial findings—treatment facilities did not have reduced mortality and morbidity after the Checklist was implemented.

Several of the time-to-complete practice measures seemed implausibly fast to be of sufficient quality. In particular, tasks like initiation of breast feeding, initiation of skin-to-skin contact, discussion or family planning and referrals likely require more time in expectation than is currently being allocated based on our study results. Referrals, which were the most time-consuming task overall, were still completed within 2 min. Given the difficulty of breastfeeding initiation,[23 24] it is unlikely that a mean task time of 24 s (SE=2 s) accurately captures the true time required to successfully initiate breast feeding. In other cases, the timing seems plausible. For example, the Centers for Disease Control and Prevention recommends handwashing for 15–20 s.[25] In our sample, handwashing took an average of 29 s. Similarly, although the self-reported sample of task-time estimates is skewed towards tasks that the BAs perceived as relatively more

burdensome, the self-reported time-to-complete tasks remained lower than expected a priori. These results are similarly indicative that time-to-complete Checklist-related practices are too low to have been consistently performed at high quality. Further research is needed to estimate minimum time requirements for the performance of practices at high quality and how the Checklist, when implemented at high quality, impacts the staffing needs of a facility. One potential downside of a checklist-based intervention is a desire to get through the items as quickly as possible rather than at the pace required to perform each task at high quality. Further quality approaches may also consider how to incorporate incentives for not just completing a checklist but reaching quantifiable quality benchmarks for the checklist items.

Although individual practices took less than 2 min to complete on average and overall less than 5 min for the full practice list, it is still possible that the workload of clinical care (and/or administrative tasks) before the introduction of the Checklist was sufficiently demanding that BAs did not have the slack to take on any incremental tasks newly introduced with the Checklist. Across all our data collection methods, however, high-quality clinical care was not the major time use of BAs in the BetterBirth study population. One of the main open questions from our time-use data collection is how to understand and estimate the time constraints faced by labour and delivery ward BAs. The nature of labour and delivery ward care requires long periods of waiting followed by high-stress, high-demand moments of clinical care. Could moments of inactivity actually be high-stress, high-alert contexts compared with times when the BA is truly on break? How should we differentiate between breaks that are necessary versus time that could be reallocated towards high-quality clinical care? How would the per cent of time spent conducting clinical care change if quality of care improved? Our data highlight the importance and difficulty of estimating supply-side constraints in the highly unpredictable context of labour and delivery wards. In the future, it will be important to continue to estimate how quality improvement interventions impact the time use of providers including work to parse out time which appears to be free, but in reality may be a version of alert waiting.

Ensuring quality care at facilities requires thoughtful clinical care practices and staffing strategies.[26–30] Our data collection efforts add empirical evidence on how BAs in Uttar Pradesh, India, use their time across both clinical and non-clinical care under varying levels of patient demand. In future implementations of the Checklist, our data on the time-to-complete clinical tasks as well as the time use of BAs can serve as both a model of how to collect data and as a baseline for potential data collection improvements that could address lingering questions raised in this paper.

There are several limitations in our methods and data collection. Although we began with separate categories for breaks and downtime, this distinction was not clear during the actual observation. We cannot reliably distinguish true breaks from watchful waiting. Practices were meant to be timed from start to finish, pausing for breaks. For instance, if a family planning discussion began but was interrupted by breastfeeding initiation, the stopwatch should have been stopped and restarted when the family planning discussion restarted to capture the overall time required for that practice. Given the consistently short task-time estimates, this may not have occurred. In our survey of BAs, our sample size is relatively small (n=83), the responses may not generalise to the broader BA population in our study and Uttar Pradesh more broadly. Instead of asking the BAs to estimate the task time for all 18 practices, we only asked for the top three in an effort to keep the survey short. However, it limits our self-reported task times to only those activities that BAs considered especially time consuming, biasing the self-reported results upwards. Finally, this study was not designed to study variation in BA time use by facility type, a stratified analysis by facility type may help explain some of the variation in patient load per BA and BA time use.

There are often calls for measurable indicators of healthcare quality. In the recent Lancet Global Health Commission on High-Quality Health Systems, many of the available quality metrics rely on the proportion or number of evidence-based practices performed.[9] Although completion of tasks is important, our evidence suggests simply performing evidence-based care does not itself ensure quality. This outcome mirrors the message that coverage of services does not equate to quality. When future quality improvement and evidence-based care interventions are implemented, it will remain important to understand how the intervention fits within the broader responsibilities and time demands of BAs as well as estimating time demands by facility type. Quality care requires essential care is completed at a satisfactory level beyond simple completion of tasks.

**Author affiliations**
[1]Interfaculty Initiative in Health Policy, Harvard University, Cambridge, Massachusetts, USA
[2]Ariadne Labs, Harvard T H Chan School of Public Health/Brigham and Women's Hospital, Boston, Massachusetts, USA
[3]Population Services International, Lucknow, India
[4]Community Empowerment Lab, Lucknow, India
[5]Division of Hematology/Oncology, Department of Medicine, University of California San Francisco, San Francisco, California, USA
[6]Piramal Swasthya Management and Research Institute, Hyderabad, India
[7]Center for Health Decision Science, Harvard University T H Chan School of Public Health, Boston, Massachusetts, USA
[8]Division of Global Health Equity, Brigham and Women's Hospital, Boston, Massachusetts, USA
[9]Department of Medicine, Harvard Medical School, Boston, Massachusetts, USA

**Acknowledgements** We recognise the governments of India and Uttar Pradesh for collaboration and support to conduct this trial in public health facilities. We thank the facility staff, women and newborns for their participation in the study. We are indebted to Sriya Srikrishnan, Ezinne Eze-Ajoku, Bharath Kumar, Krishna Kumar, the late Dr Narender Sharma and all the data collectors on the facility-based data associate team. We are grateful to the members of the trial's scientific advisory

committee who contributed crucial guidance to the development of this study protocol.

**Contributors**  KEAS, SCR and KTL conceived the study. KTL conducted the analysis. VPS and TK supervised the field operations and reviewed the survey instruments. KTL, AK, KEAS and LB wrote the manuscript. AK provided extensive edits and manuscript support. LB, DET, MMD and AJ provided operational support for the study. MR provided clinical inputs for the study. All authors (KEAS, LB, DET, VPS, MMD, AJ, MR, TK, AK, SCR and KTL) reviewed the manuscript and provided edits. KTL accepts full responsibility for the finished work and/or the conduct of the study, had access to the data, and controlled the decision to publish.

**Funding**  This study was funded by Bill & Melinda Gates Foundation (OPP1017378).

**Competing interests**  None declared.

**Patient consent for publication**  Not required.

**Ethics approval**  This study involves human participants. The study protocol was approved by the ethics committees of Community Empowerment Lab (Ref No 2014006), Jawaharlal Nehru Medical College (Ref No MDC/IECHSR/2015-16/A-53), Harvard T H Chan School of Public Health (Protocol 21975-102), Population Services International (Protocol ID: 47·2012), WHO (Protocol ID: RPC 501) and Indian Council of Medical Research. The protocol was reviewed and reapproved on an annual basis. We obtained consent from each facility's leadership for trial participation and data collection on eligible mothers from facility registers. Birth attendants and facility staff verbally agreed to participate prior to trial initiation. Independent observers obtained written consent from women or their surrogates and verbal consent from birth attendants prior to observation. Participants gave informed consent to participate in the study before taking part.

**Provenance and peer review**  Not commissioned; externally peer reviewed.

**Data availability statement**  Data are available in a public, open access repository. Data are available on the Harvard Dataverse platform under the BetterBirth Dataverse website. This can be found at https://dataverse.harvard.edu/dataverse/BetterBirthData.

**ORCID iDs**
Katherine T Lofgren http://orcid.org/0000-0002-3346-5504
Stephen Charles Resch http://orcid.org/0000-0002-0858-5467
Katherine E A Semrau http://orcid.org/0000-0002-8360-1391

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
