## [Reviewer comments · BMJ Open]

ARTICLE DETAILS

TITLE (PROVISIONAL)	Estimating maternity ward birth attendant time use in India: A microcosting study
AUTHORS	Lofgren, Katherine; Bobanski, Lauren; Tuller, Danielle; Singh, Vinay P.; Marx Delaney, Megan; Jurczak, Amanda; Ragavan, Meera; Kalita, Tapan; Karlage, Ami; Resch, Stephen; Semrau, Katherine

VERSION 1 – REVIEW

REVIEWER	AMOAKOH-COLEMAN, MARY University of Ghana
REVIEW RETURNED	09-Jul-2021

GENERAL COMMENTS	Well written paper and addressing a relevant issue especially in LMIC where human resource challenges have been clearly identified as impacting on quality of care, Understanding some of these nuances of what goes into service provision will inform capacity building, task shifting and generally human resource distribution among others. 1. Did the results shown differ per the type of facility or not? Are the findings similar at primary level facility and referral center for example? Since these different facility types were used, it will be useful to know if the time use varies across them or are similar.
--

REVIEWER	Hirose, Atsumi Karolinska Institutet, Department of Global Public Health Sciences
REVIEW RETURNED	09-Jul-2021

GENERAL COMMENTS	This study contains interesting and perhaps useful data to understand the time demand on facility-based birth attendants in carrying out essential clinical practices in India. Although study results are mainly descriptive without any control or comparison group, data may be useful for those who are investigating the intersection between improving quality of delivery care/service delivery and (already strained) health workforce in LMICs. The presentation of results needs to be improved. Results are mostly presented in graphs. I think some of them could have been presented better in a table with a clear sample size (N). Abstract: Setting: I can understand why the authors stated ‘it was a matched-pair, cluster-randomized, controlled trial to test...’ but it almost gives the wrong impression that this particular study was a trial, which is not. Perhaps it is not necessary to state the study
---

	design of the larger trial in which this particular study was nested within, to avoid confusion. Participants: not clear how many birth attendants were included. N provided here is the number of facilities, not the number of participants included in the survey. Results: the primary outcome (% of staff time required to complete the practices) is not reported here (though it should be) but reported in the conclusions. Main text Introduction: Page 6 para 2 starting with “To study the effect of...” This paragraph is not about this particular study. The focus of this paper should not be on the larger trial but this smaller sub-study. I would suggest to revise this para to clarify the relationship of this particular study with the larger trial. (I.e focus on the relationship rather than the study design, aim etc of the larger trial which is not particularly relevant for this paper). Methods: Page 7 last para “Time use data collection”. It would be helpful if a short description of the three data collection methods was added at first mention. I couldn’t immediately get that the next three paragraphs were about the three methods (as the sub-headings are not clear). Page 8, under ‘perceived time-demand of the checklist by BAS’, key information is missing. The timing of the survey? Eligibility criteria for the participants? The number of participants included in the survey? How the participants sampled? Page 9, under “BA time use in the labour and delivery ward” key information is again missing. How many hours were sampled? How did you determine the sample size? Results: It would be helpful to present an overview /description of the samples for this study, perhaps at the beginning of the results section using a table? As it is, it’s hard to see the overall picture as there is no summary table of the samples. N is provided in the text in the first para, but I can’t see how they distribute across the 18 practices without a table as the Ns do not add up to 1559. In the abstract, I read that the primary outcome measure is the percent of staff time required to complete the essential birth practices. This metric was reported finally in the last sub-section under “BA time use in the labour and delivery wards”. I wonder why data collected from the first data collection method (time demand of checklist practice) was not used to calculate the primary outcome? Seems to me it will produce a better estimate as the duration of each task was measured? Under “Perceived time-demand of the checklist by BAs”, “We directly asked Bas...” is a repeat of the method section? Under “BA time use in the labour and delivery ward”, it is not clear how many ‘work samples’ were observed. Please could you show results in a table? Currently, % are only presented in the text, without telling how many samples were observed (i.e the denominator and the numerator are not shown at all). The total sample size (N) by types of tasks carried out should be presented. I wonder what explains the wide range (0-8) of patients per BA? The time of the day does not seem to influence this (which also makes me wonder why the authors decided to show the number of
--	--

	patient per BA by time which does not explain the variation?) What about type of facilities/level of care? In the last paragraph of this sub-section, the relationship between patient load and the proportion of 'down time' is 'analysed' but because we do not know what determines BA having 3 or more or no patient, it is difficult to understand whether there is a "third" factor /confounder involved. Figure 2- What does the length /width of each of the coloured bars represent? Figure 3 – Sample size? Discussion The first para line 49. When I read this sentence, I nearly lost confidence in this study as it gives the impression that even the authors do not believe in what they did. 'Ambiguously measured'? Issues with the face validity? These are perhaps limitations of the study and could be discussed elsewhere. Are there similar studies to compare results to? Results are hardly discussed in the light of existing literature, in particular, patient-load vs time use is hardly discussed in relation to existing literature.
--	--

VERSION 1 – AUTHOR RESPONSE

Reviewer: 1

Miss MARY AMOAKOH-COLEMAN, University of Ghana

Comments to the Author:

Well written paper and addressing a relevant issue especially in LMIC where human resource challenges have been clearly identified as impacting on quality of care, Understanding some of these nuances of what goes into service provision will inform capacity building, task shifting and generally human resource distribution among others.

1. Did the results shown differ per the type of facility or not? Are the findings similar at primary level facility and referral center for example? Since these different facility types were used, it will be useful to know if the time use varies across them or are similar.

Response:

Thank you for this question. Unfortunately, although the overall BetterBirth study has a sample of 120 facilities, the surveys included in this study did not include sufficient samples across different types of facilities for a stratified breakdown to be possible consistently across all survey types. We appreciate that this is an important question and have added a sentence suggesting future work look at facility-type breakdowns into the discussion. Here is a quotation of what we've added:

"When future quality-improvement and evidence-based care interventions are implemented, it will remain important to understand how the intervention fits within the broader responsibilities and time demands of BAs as well as estimating time demands by facility-type."

Reviewer: 2

Dr. Atsumi Hirose, Karolinska Institutet, Imperial College London

Comments to the Author:

This study contains interesting and perhaps useful data to understand the time demand on facility-based birth attendants in carrying out essential clinical practices in India. Although study results are mainly descriptive without any control or comparison group, data may be useful for those who are investigating the intersection between improving quality of delivery care/service delivery and (already

strained) health workforce in LMICs.

The presentation of results needs to be improved. Results are mostly presented in graphs. I think some of them could have been presented better in a table with a clear sample size (N).

Response:

Thank you for your suggestion. We have added supplemental tables to the appendix which are detailed below based on your specific comments and concerns.

Abstract:

Setting: I can understand why the authors stated 'it was a matched-pair, cluster-randomized, controlled trial to test...' but it almost gives the wrong impression that this particular study was a trial, which is not. Perhaps it is not necessary to state the study design of the larger trial in which this particular study was nested within, to avoid confusion.

Response:

We appreciate this feedback. We have added wording to make it clear that this was survey work within a RCT. We do want to mention the RCT as the context in which we were assessing birth attendant time use. Below is the changed wording to the abstract:

"We collected detailed survey data on birth attendant time-use within the BetterBirth trial in Uttar Pradesh, India. The BetterBirth trial tested whether the peer-coaching-based implementation of the WHO Safe Childbirth Checklist was effective in improving the quality of facility-based childbirth care. This paper details three birth attendant time-use survey strategies administered with the BetterBirth trial."

Participants: not clear how many birth attendants were included. N provided here is the number of facilities, not the number of participants included in the survey.

Results: the primary outcome (% of staff time required to complete the practices) is not reported here (though it should be) but reported in the conclusions.

Response:

Thank you for this feedback. We have added wording to make it clear both the facility and observations for each survey as a part of the abstract. Below is the changed wording to the abstract:

"We collected measurements of time-to-completion for 18 essential birth practices from July 2016 through October 2016 across 10 facilities in 5 districts (1559 total timed observations). An anonymous survey asked about the impact of the WHO Safe Childbirth Checklist on birth attendants at every intervention facility (15 facilities, 83 respondents) in the Lucknow hub. Additionally, data collectors visited facilities to conduct a census of patients and birth attendants across 20 facilities in 7 districts between June 2016 to November 2016 (610 2-hour facility observations)."

Main text

Introduction:

Page 6 para 2 starting with "To study the effect of..." This paragraph is not about this particular study. The focus of this paper should not be on the larger trial but this smaller sub-study. I would suggest to revise this para to clarify the relationship of this particular study with the larger trial. (I.e focus on the relationship rather than the study design, aim etc of the larger trial which is not particularly relevant for this paper).

Response:

Your point is taken. We have deleted this paragraph and instead included the larger trial as an introductory sentence to the paragraph below to reflect your suggestion. Below is the changed wording at the start of the subsequent paragraph:

"The BetterBirth trial was a matched-pair, cluster-randomized, controlled trial of a coaching-based implementation of the Checklist in Uttar-Pradesh, India to test the effect of the intervention on a composite outcome of perinatal mortality, maternal mortality, or maternal severe complication within 7

days of giving birth.[19,20] As part of the BetterBirth trial, we conducted data collection to measure the time-demands of the Checklist practices, with the primary intent of informing a cost-effectiveness analysis (CEA) of the BetterBirth trial.”

Methods:

Page 7 last para “Time use data collection”. It would be helpful if a short description of the three data collection methods was added at first mention. I couldn’t immediately get that the next three paragraphs were about the three methods (as the sub-headings are not clear).

Response:

This is helpful feedback. We took out extra information about the overall trial, took out a heading where the intro to the subsections was placed, and added some references to the subsections before the subsections start. Below is the new introduction to the methods section:

“This paper details three time-use data collection methods to triangulate the time-burden of the Checklist practices within the broader time-demands on BAs within the BetterBirth trial. Data collectors (N=16) were junior nurses who received training and supportive supervision for data quality assurance across all three data collection methods (each described in more detail in subsections below). We captured 18 specific Checklist practices (Appendix Table A1) as well as non-Checklist clinical care, administrative duties, and break/downtime. Although the intention was to distinguish between a scheduled break and non-scheduled downtime, efforts to delineate between these two activities by data collectors was difficult in practice. For the purposes of this paper, ‘downtime’ refers to a mix of scheduled breaks as well as idle time for other reasons, such as no patients or watchful waiting during clinical care. We first measured time-to-completion for 18 practices via direct BA observation during clinical practice (time-demand). We then surveyed BAs about their experience during the BetterBirth trial (perceived time-demand). Finally, we visited facilities and conducted both a census of births as well as observing clinical care activities at regular intervals (BA time-use).”

Page 8, under ‘perceived time-demand of the checklist by BAS’, key information is missing. The timing of the survey? Eligibility criteria for the participants? The number of participants included in the survey? How the participants sampled?

Response:

We have added respondent sample size and the timing of the survey.

“The anonymous survey asked general questions about the impact of the Checklist on the daily routines and workloads of BAs (83 respondents) at every intervention facility (15 facilities) in the Lucknow hub (the cost-effectiveness data collection survey region with 30 total facilities) from June to July 2016. All staff working at the facility on the day of data collection were provided their survey and could answer anonymously”

Page 9, under “BA time use in the labour and delivery ward” key information is again missing. How many hours were sampled? How did you determine the sample size?

Response:

We have added facility-hours sampled

“610 2-hour facility observations were taken across 20 facilities in 7 districts from June 2016 to November 2016.”

Results:

It would be helpful to present an overview /description of the samples for this study, perhaps at the beginning of the results section using a table? As it is, it’s hard to see the overall picture as there is no summary table of the samples.

Response:

We appreciate your interest in more detailed sample size information. However, the three data collection efforts are very different in purpose, method of collection, and sample size unit. In some

cases it's time measurements, in others it's respondents, and in others it's a mix of facility census data on number of women giving birth per hour and BA observed facility hours. These differences are why we've tried to create clear, concise methods and results sections that mirror each other and include repetitive information about the sample sizes of the specific data discussed in each section. We don't think a table trying to combine this information will help and may instead create further confusion. The methods are complimentary, but not similar. Instead, we have added more tables to the appendix to address your concern (see responses below for more details).

N is provided in the text in the first para, but I can't see how they distribute across the 18 practices without a table as the Ns do not add up to 1559.

Response: The sample size for each measure is included in Appendix Table A6. The sum of the directly measured sample sizes listed in that table is 1559. Only a few specific measures are called out in results section. We have added a reference to the appendix table for clarity.

"Across all Checklist practices, a total of 1,559 practices were directly timed from 35 unique birth attendants (BAs) across 10 facilities during clinical care (see Appendix Table A6 for practice-specific sample sizes)."

In the abstract, I read that the primary outcome measure is the percent of staff time required to complete the essential birth practices. This metric was reported finally in the last sub-section under "BA time use in the labour and delivery wards". I wonder why data collected from the first data collection method (time demand of checklist practice) was not used to calculate the primary outcome? Seems to me it will produce a better estimate as the duration of each task was measured?

Response:

The direct times give us a sense for how long it takes to complete a specific task but not how frequently that task is done in clinical practice. The last section describes the direct observation of what birth attendants spend their time doing in clinical practice.

Under "Perceived time-demand of the checklist by BAs", "We directly asked Bas..." is a repeat of the method section?

Response:

Have changed this sentence to say:

"Respondents were asked to rank the three most time-consuming Checklist practices and estimate the time required to complete those three tasks."

Under "BA time use in the labour and delivery ward", it is not clear how many 'work samples' were observed. Please could you show results in a table? Currently, % are only presented in the text, without telling how many samples were observed (i.e the denominator and the numerator are not shown at all). The total sample size (N) by types of tasks carried out should be presented.

I wonder what explains the wide range (0-8) of patients per BA? The time of the day does not seem to influence this (which also makes me wonder why the authors decided to show the number of patient per BA by time which does not explain the variation?) What about type of facilities/level of care?

Response:

Have added a table to the appendix (Appendix Table A8).

There is lots of heterogeneity in the number of women giving birth at a given time, that is one of the reasons staffing is so hard in this clinical context.

We showed number of patients per BA by time because we did not know a priori that it wouldn't matter which makes the finding helpful context for both us and future researchers

Facility/levels of care where unfortunately outside the scope of this study

In the last paragraph of this sub-section, the relationship between patient load and the proportion of 'down time' is 'analysed' but because we do not know what determines BA having 3 or more or no

patient, it is difficult to understand whether there is a “third” factor /confounder involved.

Response:

Agree, this is descriptive data and confounders likely exist. The reasons for BA having more than 3 patients compared to none is not possible to rigorously assess in this study design. In labor, there is known stochastic variability, which makes labor and delivery wards difficult staffing contexts. Waiting until daylight is one potential source of variation, which is why we included a breakdown by hour.

Figure 2- What does the length /width of each of the coloured bars represent?

Response:

Figure 2 shows the distribution birth attendant time use stratified by patient load. The middle black line is the median and the full colored bar is the interquartile range. We’ve added a legend to this effect for the Figure.

Also added the text:

“Sample size breakdowns for individual work sampling observations by patient load and task-type are available in Appendix Table A8.”

Figure 3 – Sample size?

Response:

Figure 3 is a distributional graph. The sample size is the sum of the y-axis values. We prefer visual displays of information, but we appreciate that you respond better to table-based results. As such, we have added a complimentary sample size table to the appendix.

Time Demand (stopwatch direct measures)

- Appendix Table A6 (existing)

Perceived Time Demand (self-report from BA)

- Appendix Table A6 (existing)

- Appendix Table A7 (existing)

Time Use (birth census + work sampling)

- Appendix Table A8 (new)

- Appendix Table A9 (new)

Discussion

The first para line 49. When I read this sentence, I nearly lost confidence in this study as it gives the impression that even the authors do not believe in what they did. ‘Ambiguously measured’? Issues with the face validity? These are perhaps limitations of the study and could be discussed elsewhere.

Response:

We believe strongly that this is an important contribution to the field and also has lessons to learn from – a better measurement of watchful waiting vs. real downtime being one of them. Have changed wording to focus on the importance of the work and lessons to learn from:

“Further, our results show a high proportion of ambiguously measured downtime, a lesson to learn from in future studies to differentiate watchful waiting from true downtime”

Are there similar studies to compare results to? Results are hardly discussed in the light of existing literature, in particular, patient-load vs time use is hardly discussed in relation to existing literature.

Response:

If the reviewer has specific studies in mind, we would be happy to consider them in the context of this study. However, we found the relevant literature to be sparse. For instance, in trying to vet the timed clinical practices, finding citations that supported quality time-to-complete tasks was missing for almost all practices (with handwashing as an exception).

VERSION 2 – REVIEW

REVIEWER	Hirose, Atsumi Karolinska Institutet, Department of Global Public Health Sciences
REVIEW RETURNED	20-Oct-2021

GENERAL COMMENTS	Thank you for addressing most of my queries. I still think that the primary outcome should be reported in the results paragraph rather than the conclusion paragraph of the abstract. (i.e "On average, WHO Checklist clinical care accounted for less than 7% of birth attendant time-use per hour" should belong to the results?) In relation to the above, I read in the main text that "At 3 or more patients-per-BA, the Checklist accounted for 7% of BA time (out of a total 24% of the hour spent on clinical care)" but i'm not sure if the authors reported the proportion of hour spent on completing the checklist when the patient load was less than 3? I may have missed it but this should be reported in the main text and not just in the abstract. It is pity that the number of the facilities included in the study does not allow stratification of the patient load per BA (and other outcomes) by types of facilities, which the first reviewer queried. Perhaps this is one of the important 'third' factors that may partly explain the variability of the patient load per BA. The authors may want to acknowledge this as a limitation. I don't have any specific study in mind when I asked to compare to existing literature. I thought there might be some time motion studies.
---

VERSION 2 – AUTHOR RESPONSE

Reviewer: 1

I still think that the primary outcome should be reported in the results paragraph rather than the conclusion paragraph of the abstract. (i.e "On average, WHO Checklist clinical care accounted for less than 7% of birth attendant time-use per hour" should belong to the results?)

Response:

Thank you. We have moved to address your concern and have made minor changes to the abstract to add an additional sentence to the conclusion while staying under the 300 word limit. Here is the revised conclusion:

"We did not find that a coaching-based implementation of the WHO Checklist was a burden on birth attendant's time-use. However, questions remain regarding the performance quality of practices and how to accurately capture and interpret idle and break time."

In relation to the above, I read in the main text that "At 3 or more patients-per-BA, the Checklist accounted for 7% of BA time (out of a total 24% of the hour spent on clinical care)" but i'm not sure if the authors reported the proportion of hour spent on completing the checklist when the patient load was less than 3? I may have missed it but this should be reported in the main text and not just in the abstract.

Response:

Thanks for bringing this point of confusion up. The abstract is worded with the phrase “less than 7%” because we are reporting the upper bound there (the same number as the 3 or more patients-per-BA in the paragraph referred to here). We have changed this paragraph to include the percent of time spent on WHO Checklist practices for each patient-load bin to clarify. Here is the revised wording:

“When there were no patients, BAs spent the majority of their time in downtime (80% of time) or conducting administrative tasks (15% of time) and less than 1% of time on Checklist clinical care. Once the patient-load increased to 1-2 patients-per-BA, BA time-use shifted towards clinical care (23% of BA time; 5% Checklist specific) as well as administrative tasks (26% up from 15% with no patients). At 3 or more patients-per-BA, the Checklist accounted for 7% of BA time (out of a total 24% of the hour spent on clinical care).”

It is pity that the number of the facilities included in the study does not allow stratification of the patient load per BA (and other outcomes) by types of facilities, which the first reviewer queried. Perhaps this is one of the important 'third' factors that may partly explain the variability of the patient load per BA. The authors may want to acknowledge this as a limitation.

Response:

Thank you. We added a sentence after the first review round to address the reviewers comment. Here is that quote:

“When future quality-improvement and evidence-based care interventions are implemented, it will remain important to understand how the intervention fits within the broader responsibilities and time demands of BAs as well as estimating time demands by facility-type.”

In addition to specifically address your concerns this round we have added a bullet to the summary strengths and limitations section calling this out and have added the following wording to the manuscript (page 15):

“Finally, this study was not designed to study variation in birth attendant time-use by facility-type, a stratified analysis by facility-type may help explain some of the variation in patient load per birth attendant and birth attendant time-use.”

I don't have any specific study in mind when I asked to compare to existing literature. I thought there might be some time motion studies.

Response:

Thanks for clarifying. We have added a time motion and work sampling references based on this clarification that we did consider during study design. We are not aware of any applied studies in a similar clinical context to include as references.

- Finkler et al. 1993 “A Comparison of Work-Sampling and Time-and-Motion Techniques for Studies in Health Services Research”

You can find reference to the papers above in the paper on page 6 at first mention of the methods approach:

- “This paper details three time-use data collection methods to triangulate the time-burden of the Checklist practices within the broader time-demands on BAs within the BetterBirth trial [21].”

There is also another existing reference on the technique (was citation 21, now 22) titled “A technical note on using work sampling to estimate the effort on activities under activity-based costing.”

We wish there was more depth of applied research in the maternity ward context to include as citations. However, the use of these 3 methods in the context of maternity ward time-use is one of the major contributions of this study.